# Cognitive Biases and Addictive Disorders: A Bibliometric Review

## Melvyn W. B. Zhang

Family Medicine and Primary Care (FMPC), Lee Kong Chian School of Medicine,
Nanyang Technological University, Clinical Sciences Building, 11 Mandalay Road, Singapore 308232, Singapore;
melvynzhangweibin@gmail.com

**Abstract:** Introduction: Since the early 2000s, there have been extensive investigations into cognitive biases in addictive disorders. The advances in the field have led to the discovery that cognitive bias exists in substance disorders and could in turn be modified. To date, there have been primary studies and meta-analysis demonstrating the existence of these biases and the effectiveness of cognitive bias modification (i.e., whereby such biases are retrained). There remains a lack of understanding of how the field has progressed and the research gaps, in light of the evidences provided by these primary studies. Objectives: A bibliometric analysis of the publications to date was performed to provide a map of the work that has been done so far. This would help researchers to better understand the development of cognitive bias research, the direction of the research, and the recent trends. Methods: For the purposes of this bibliometric research, Web of Science (WOS) was used in the identification of relevant articles. To identify the relevant articles, the following search strategy was implemented, that of ((((((TS = ("cognitive bias")) OR TS = ("attention bias")) OR TS = ("approach bias")) OR TS = ("avoidance bias)) OR TS = ("interpretative bias"))). Bibliometric data analysis was conducted based on the identified articles. Results: A total of 161 citations were eventually included. These citations were published between 1994 and 2022. The average number of citations per documents was 26.73. Of these 161 citations, 122 were articles, 2 were editorials, 3 were corrections to the original manuscript, 5 were reviews, and 29 were meeting abstracts. The analysis of the trend of topics has shown that researchers were focused on understanding and gaining insights into cognitive biases and potentially examining the association between cognitive biases and cravings and aggression in the early days. Over the years, there has been an evolution into examining specific unconscious biases, namely, that of attention and approach biases. In the most recent years, the investigations have been more focused on examining bias modification/retraining. Conclusions: From our knowledge, this is the first bibliometric analysis that has been undertaken to explore all the publications related to cognitive bias in the field of addiction. The insights gained from this article could inform future research.

**Keywords:** cognitive bias; approach bias; attention bias; addiction; bibliometric

## 1. Introduction

McCusker (2001) [1] highlighted how the understanding of addictive disorders should move away from conventional theoretical approaches, recognising that cognitive biases might result in individuals selectively processing substance-related stimuli in their environment. McCusker (2001) also reported how these processes could motivate addictive behaviours and how these processes occur at an automatic and implicit level that individuals are unaware of [1]. Since then, there have been tremendous advances in the field involving the examination of cognitive biases in addictive disorders and more recently, further exploration into the retraining or modification of these automatic, unconscious processes. Some of the cognitive biases present amongst individuals with addictive disorders include attention and approach/avoidance biases [2,3]. Attention biases refer to how one's attentional processes are preferentially allocated towards substance-related cues in the environment [2,3]. Approach biases refer to the automatic behavioural tendencies for

individuals to reach out towards substance-related cues in their natural environment [2,3]. Studies have reported how these biases could result in lapses and relapses amongst individuals with addictive disorders and, hence, retraining or bias modification is crucial [4].

There has been a series of systematic reviews and meta-analyses published that provided evidence for the existence of cognitive biases in addictive disorders and the effectiveness of bias modification. For example, Zhang et al. (2018), in their review of cognitive biases in opioid, cannabis, and stimulant use disorders, reported that biases are present in these disorders [5]. Their work was furthered by other work, such as that of Maclean et al. [6], which provided further justification for the existence of such biases amongst individuals with opioid use disorders. Others, such as Cristea et al. (2016), examined the effectiveness of cognitive bias modification for addictive disorders, such as smoking and alcohol use [2], and Boffo et al. (2019) [7] evaluated the effect sizes for cognitive bias modification and reported that bias modification has had a small effect on cognitive biases and their relapse rates. From our understanding, there remains to date a recent review by Gober et al. (2020) that attempted to map out the progress in the field of cognitive biases. The authors mapped out the articles in which cognitive bias modification has been applied and some of the data in each area [8].

The previous work has helped researchers map out interventions for cognitive biases. However, the reviews were performed cross-sectionally. It is thus timely for this bibliometric review to map out the work that has been done so far and highlight some of the key publications in the field. This review aims to help researchers better understand the direction of research in the field and identify key research gaps.

## 2. Methods

The Web of Science (WOS) database was used to identify relevant articles for this bibliometric review. For most bibliometric reviews, the WOS database was utilised, given that it is superior to Scopus or PubMED/MEDLINE for several reasons [9,10]. The WOS database allows researchers to extract articles with all the necessary information, such as of titles, name of authors, total citations, and the overall download rates. Moreover, the database is comprehensive and includes citations since the 1900s and articles from all the leading high impact factor journals that are available globally.

To identify articles that were relevant to the intended topic, the following search strategy was implemented, that of ((((((TS = ("cognitive bias")) OR TS = ("attention bias")) OR TS = ("approach bias")) OR TS = ("avoidance bias)) OR TS = ("interpretative bias"))). Based on this strategy, a total of 5273 citations were identified. The search was refined by only identifying citations from the most relevant WOS categories. Articles were included and refined by the following categories: psychiatry, psychology clinical, neurosciences, psychology multidisciplinary, psychology experimental, psychology, behavioural sciences, psychology developmental, psychology biological, substance abuse, psychology social, and psychology educational. Articles were refined by the following categories, given that cognitive biases are usually investigated in psychology and considered part of psychological intervention. This resulted in a remaining of 3463 citations, which were further screened against their title and abstract to identify only the relevant articles. Figure 1 highlights the selection of the articles.

For analysis, the bibliometric data were analysed using the Biblioshiny package in R Studio, Version 1.4 (Via Cintia, I-80126, Naples, Department of Economics and Statistics, University of Naples Federico II, Italy) [11]. This package allowed for the analysis and visualisation of the following data:

a. Main information of the dataset
b. Annual scientific production
c. Journals that articles are published in
d. Leading authors in the field
e. Most cited global documents
f. Analysis of topic trends

g.     Thematic evolution of topics

h.     Co-occurrence network of authors' keywords

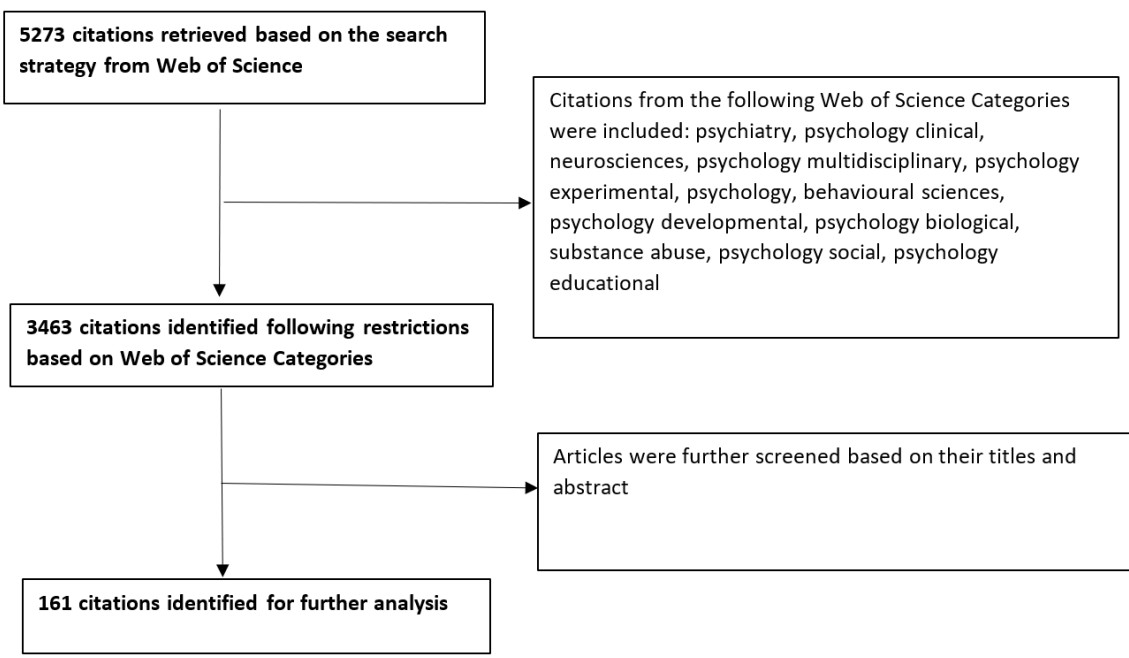

**Figure 1.** A flowchart of how studies were selected.

## 3. Results

A total of 161 citations were eventually included. With regards to the analysis pertaining to the main information of the dataset, we found that most of the citations were published between 1994 and 2022.

The average number of citations per documents was 26.73. Of these 161 citations, 122 were articles, 2 were editorials, 3 were corrections to the original manuscript, 5 were reviews, and 29 were meeting abstracts. Figure 2 provides an overview of the annual number of articles published through the years. Figure 3 shows the three-field plot (by countries, authors, and keywords).

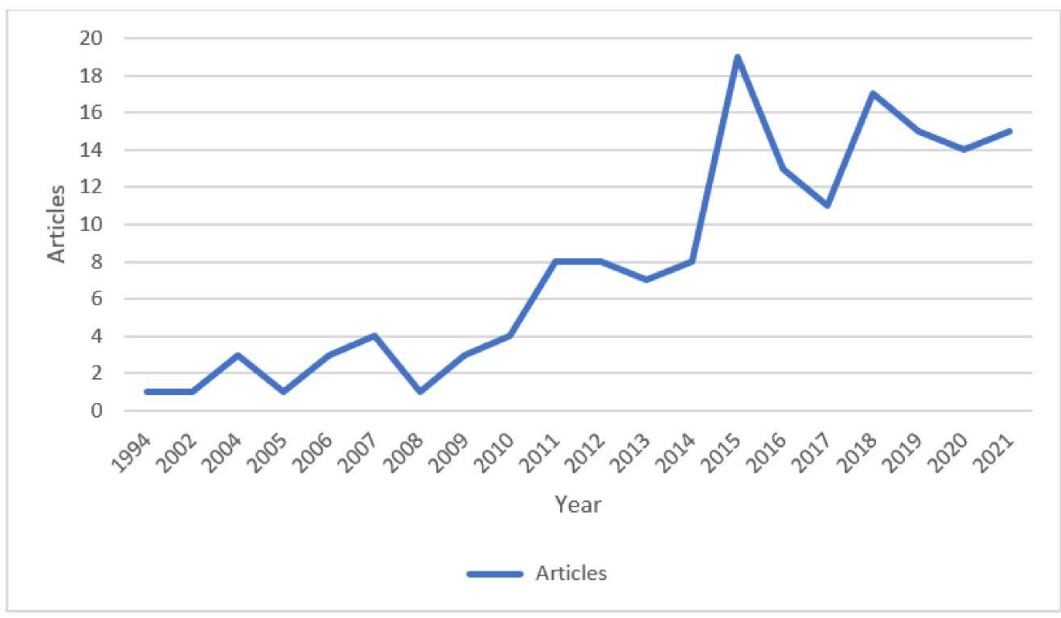

**Figure 2.** Annual scientific production of articles.

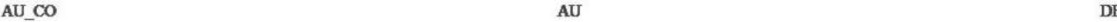

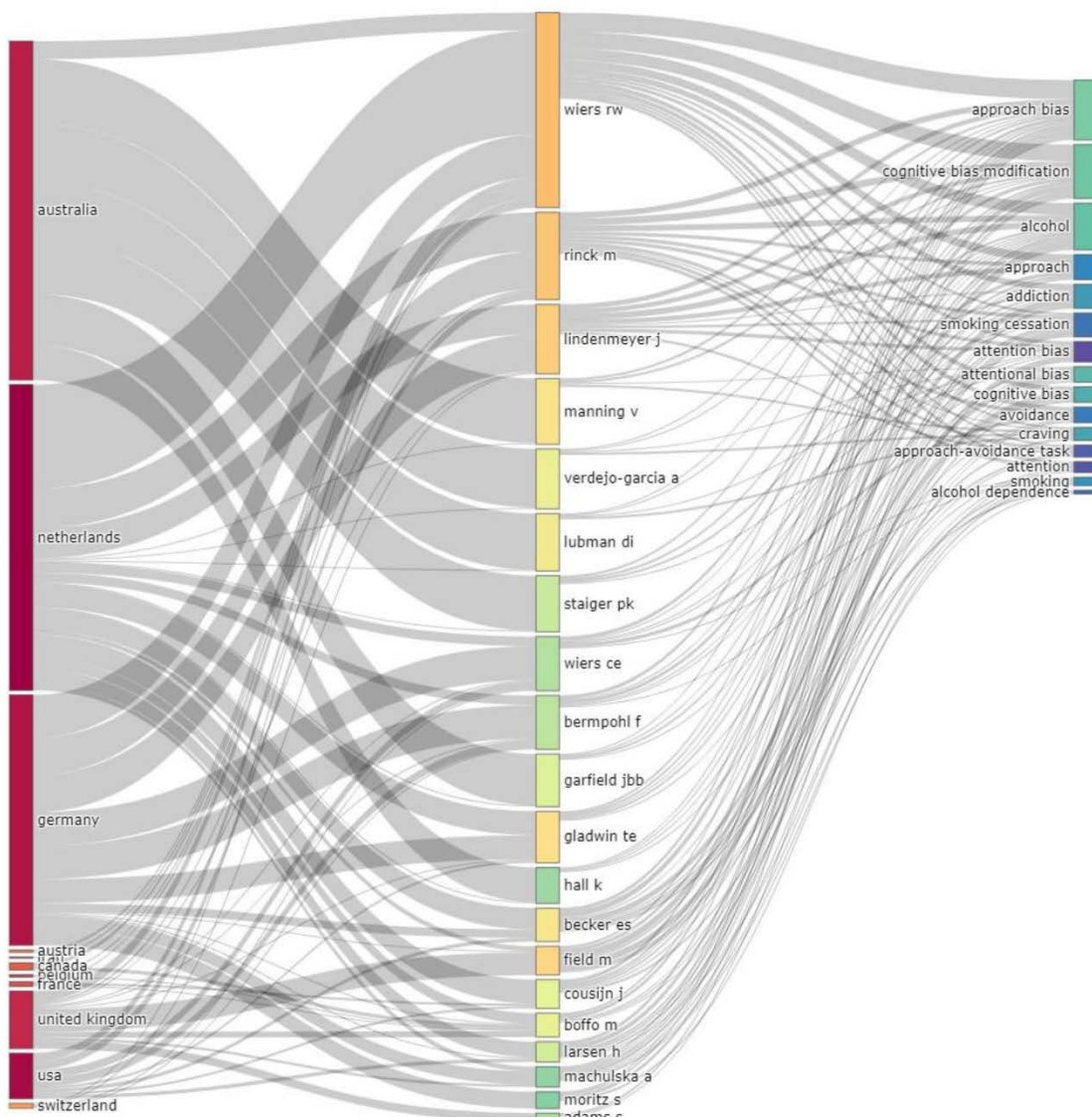

**Figure 3.** Three-field plot showing the association among countries, authors and keywords used.

The following tables illustrate the outcomes of the analysis pertaining to the journals in which articles were published and the leading authors in the field. Table 1 provides further information about the journals in which these articles were published and the leading journals in the field that published articles related to cognitive biases. Table 2 provides an overview of the leading authors in the field and the corresponding number of articles that they have published.

Table 3 provides an overview of the analysis of the most globally cited publications. This information is pertinent for us in understanding which articles have had the most impact in the field. Figure 4 illustrates the changing trends of the topics, and Figure 5 illustrates the thematic evolution of the topics over time. Figure 6 provides an overview of the co-occurrence of the authors' keywords.

**Table 1.** Journals in which articles were published.

| Journals | Articles |
| --- | --- |
| Alcoholism-Clinical and Experimental Research | 21 |
| Psychopharmacology | 14 |
| Drug and Alcohol Dependence | 10 |
| Addictive Behaviors | 7 |
| Journal of Psychopharmacology | 6 |
| Psychology of Addictive Behaviors | 6 |
| Addiction | 5 |
| Behaviour Research and Therapy | 5 |
| Experimental and Clinical Psychopharmacology | 5 |
| Alcohol and Alcoholism | 4 |
| Drug and Alcohol Review | 4 |
| European Neuropsychopharmacology | 4 |
| Journal of Behavioral Addictions | 4 |
| Addiction Biology | 3 |
| European Journal of Neuroscience | 3 |
| European Psychiatry | 3 |
| Frontiers in Psychology | 3 |
| Journal of Behavior Therapy and Experimental Psychiatry | 3 |
| Journal of Studies on Alcohol and Drugs | 3 |
| American Journal of Drug and Alcohol Abuse | 2 |

**Table 2.** Leading authors in the field.

| Authors | Articles |
| --- | --- |
| WIERS RW | 44 |
| RINCK M | 20 |
| LINDENMEYER J | 13 |
| FIELD M | 12 |
| GLADWIN TE | 10 |
| MANNING V | 10 |
| BECKER ES | 8 |
| LUBMAN DI | 8 |
| VERDEJO-GARCIA A | 8 |
| BOFFO M | 7 |
| COUSIJN J | 7 |
| GARFIELD JBB | 7 |
| LARSEN H | 7 |
| STAIGER PK | 7 |
| BERMPOHL F | 6 |
| WIERS CE | 6 |
| ADAMS S | 5 |
| HALL K | 5 |
| MACHULSKA A | 5 |
| MORITZ S | 5 |

**Table 3.** Most globally cited publications.

| Author and Journal | Publication | Total Citations | TC per Year |
|---|---|---|---|
| WIERS RW, 2011, PSYCHOL SCI [12] | Retraining Automatic Action Tendencies Changes Alcoholic Patients' Approach Bias for Alcohol and Improves Treatment Outcome | 513 | 42.75 |
| EBERL C, 2013, DEV COGN NEUROS-NETH [13] | Approach bias modification in alcohol dependence: do clinical effects replicate and for whom does it work best? | 250 | 25 |
| GRIFFITHS MD, 1994, BRIT J PSYCHOL [14] | The role of cognitive bias and skill in fruit machine gambling | 163 | 5.6207 |
| MOGG K, 2005, PSYCHOPHARMACOLOGY [15] | Attentional and approach biases for smoking cues in smokers: an investigation of competing theoretical views of addiction | 140 | 7.7778 |
| FIELD M, 2007, PSYCHOPHARMACOLOGY [16] | Experimental manipulation of attentional biases in heavy drinkers: do the effects generalise? | 138 | 8.625 |
| SCHOENMAKERS T, 2008, PSYCHOPHARMACOLOGY [17] | Effects of a low dose of alcohol on cognitive biases and craving in heavy drinkers | 137 | 9.1333 |
| COUSIJN J, 2011, ADDICTION [18] | Reaching out towards cannabis: approach-bias in heavy cannabis users predicts changes in cannabis use | 134 | 11.1667 |
| ATAYA AF, 2012, DRUG ALCOHOL DEPEN [19] | Internal reliability of measures of substance-related cognitive bias | 132 | 12 |
| FIELD M, 2006, DRUG ALCOHOL DEPEN [20] | Selective processing of cannabis cues in regular cannabis users | 118 | 6.9412 |
| WIERS CE, 2015, AM J PSYCHIAT [21] | Effects of Cognitive Bias Modification Training on Neural Alcohol Cue Reactivity in Alcohol Dependence | 101 | 12.625 |
| DUNNING JP, 2011, EUR J NEUROSCI [22] | Motivated attention to cocaine and emotional cues in abstinent and current cocaine users—an ERP study | 93 | 7.75 |
| TOWNSHEND JM, 2007, ALCOHOL CLIN EXP RES [23] | Avoidance of Alcohol-Related Stimuli in Alcohol-Dependent Inpatients | 93 | 5.8125 |
| PEETERS M, 2012, ADDICTION [24] | Automatic processes in at-risk adolescents: the role of alcohol-approach tendencies and response inhibition in drinking behavior | 90 | 8.1818 |
| WIERS RW, 2015, ADDICT BEHAV [25] | Alcohol cognitive bias modification training for problem drinkers over the web | 88 | 11 |
| WIERS CE, 2014, NEUROPSYCHOPHARMACOL [26] | Neural Correlates of Alcohol-Approach Bias in Alcohol Addiction: the Spirit is Willing but the Flesh is Weak for Spirits | 87 | 9.6667 |
| FIELD M, 2004, DRUG ALCOHOL DEPEN [27] | Cognitive bias and drug craving in recreational cannabis users | 85 | 4.4737 |
| WIERS CE, 2013, PSYCHOPHARMACOLOGY [28] | Automatic approach bias towards smoking cues is present in smokers but not in ex-smokers | 72 | 7.2 |
| EBERL C, 2014, ALCOHOL CLIN EXP RES [29] | Implementation of Approach Bias Re-Training in Alcoholism—How Many Sessions are Needed? | 71 | 7.8889 |
| CHRISTIANSEN P, 2012, PSYCHOPHARMACOLOGY [30] | Components of behavioural impulsivity and automatic cue approach predict unique variance in hazardous drinking | 68 | 6.1818 |
| MANNING V, 2016, ALCOHOL CLIN EXP RES [31] | Cognitive Bias Modification Training During Inpatient Alcohol Detoxification Reduces Early Relapse: A Randomized Controlled Trial | 66 | 9.4286 |

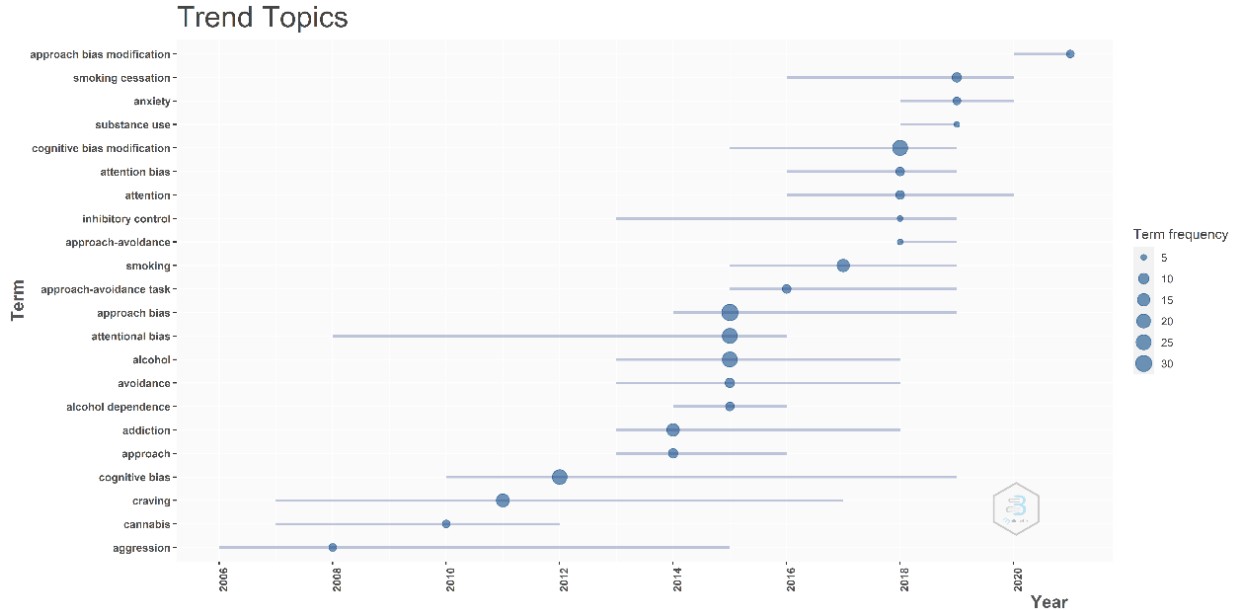

**Figure 4.** Trend of topics from 2006 to 2021.

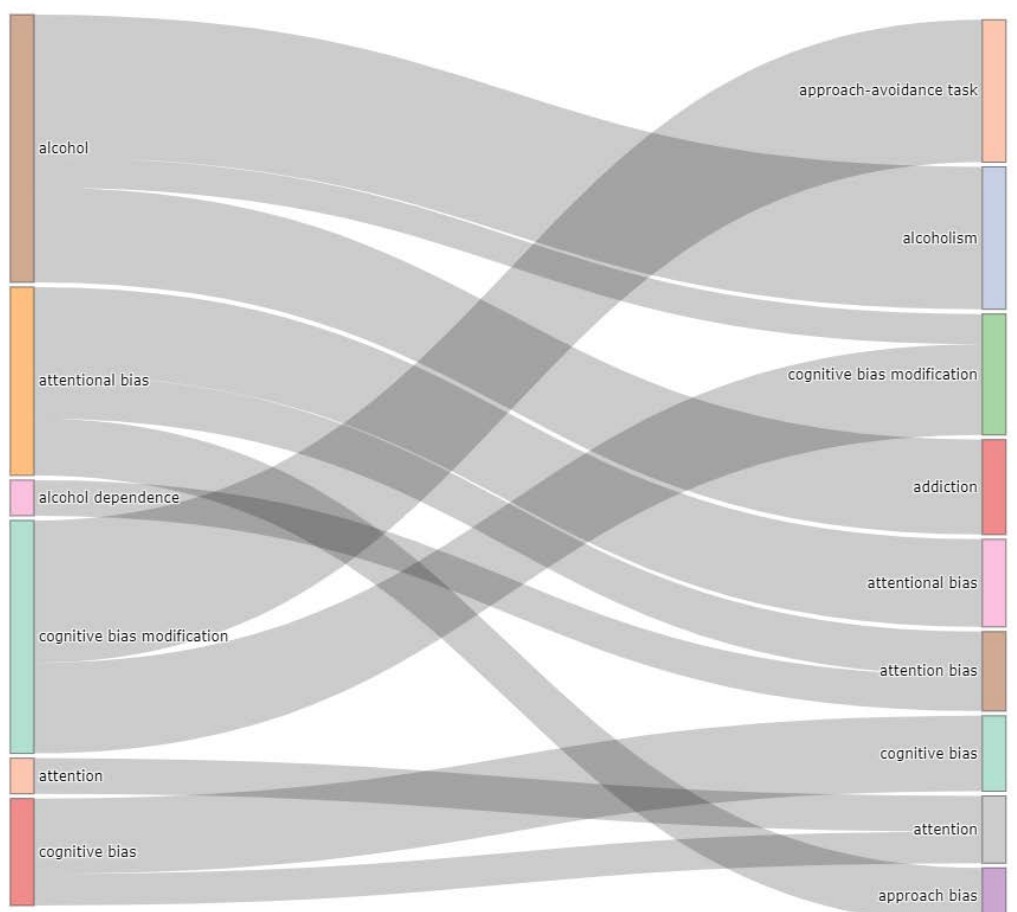

**Figure 5.** Thematic evolution of topics over the years.

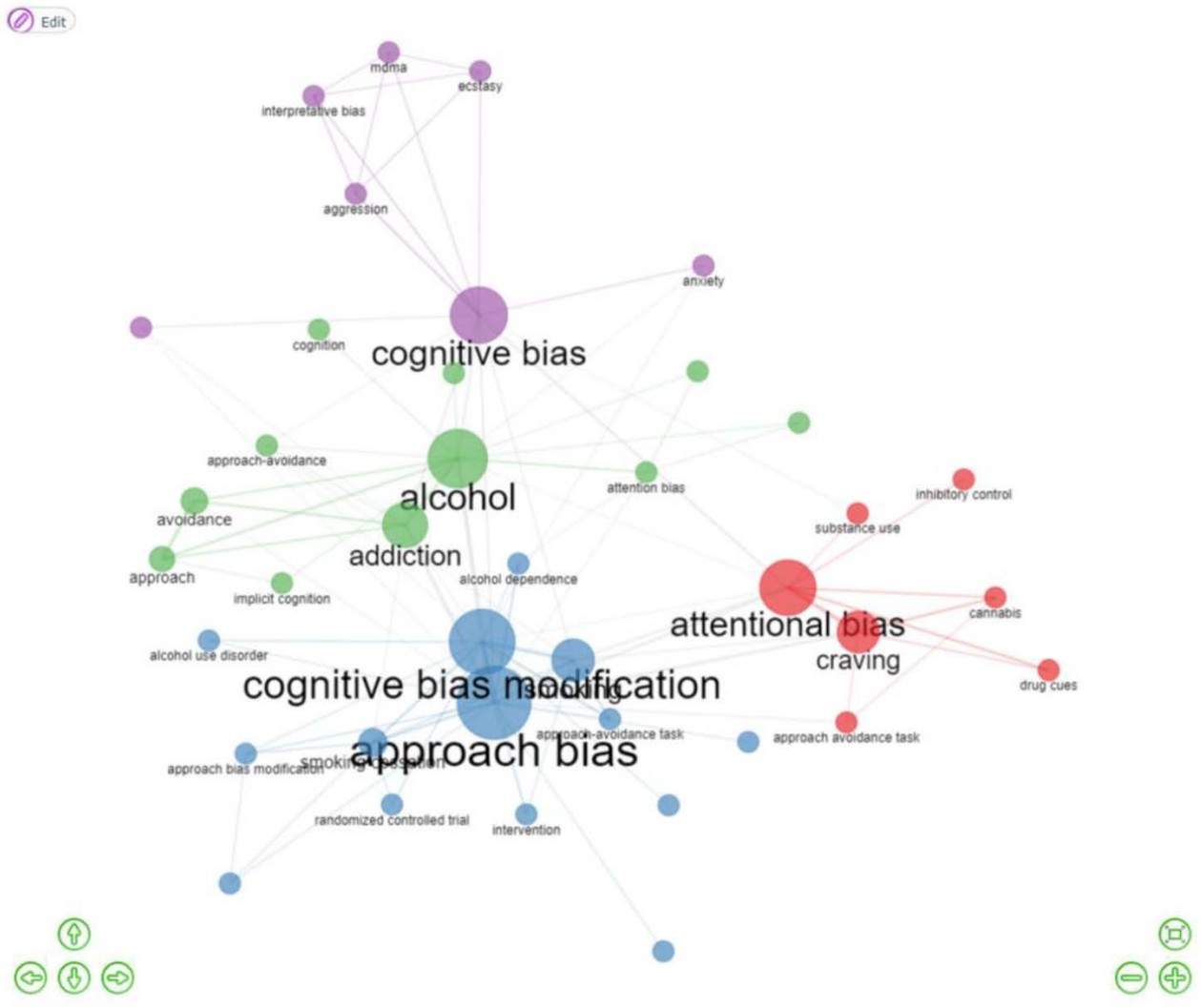

**Figure 6.** An overview of the co-occurrence of authors' keywords.

## 4. Discussion

From our knowledge, this is the first bibliometric review that has explored all the publications relating to cognitive biases for addictive disorders through the years. It is evident from this analysis that this is an area that has been well investigated, as evident by the number of articles published and cited per year. The analysis of the top authors, top globally cited articles, and the journals helped in the identification of articles that are most referred to and potentially of impact in the field. In addition, the topic trend and thematic evolution helps us understand the changing priorities of researchers and how the nature of research has evolved through the years.

As evident from the analysis of the globally most cited articles, the focus of investigation and articles that have had the most impact on the field revolves around an investigation into the following addictive disorders: alcohol, tobacco, and cannabis use disorders. Both methods of cognitive bias modification, that of attention and approach/avoidance biases modification, were considered. Some of the articles, such as that of Dunning et al. (2011) [22] and Wiers et al. (2014) [21], attempted to correlate the psychological intervention with other neuroimaging tools, such as the examination of event-related potentials. Among these articles, several articles are primarily focused on the replicability of the results following intervention. Eberl et al. (2013) [13] reported that those with alcohol use disorder, which they randomised into receiving cognitive bias modification, developed not only alcohol avoidance behaviours but have had subsequently lower rates of relapse during follow-up

at the one-year mark. However, in Field et al.'s (2007) [16] prior research, which involved the examination of generalisation attentional retraining to novel alcohol cues, they found that those who were randomised to the "avoid" alcohol group did have reduced attentional biases towards the "alcohol cues" they were presented with, but this effect failed to generalise to other cues. Eberl et al. (2014) in their article also attempted to project and determine the mean number of sessions that individuals need for approach bias retraining, should they have an underlying alcohol use disorder. The authors recommended an average of six sessions [29]. The analysis of the topmost cited articles also revealed that investigations have looked at harnessing web technologies for the delivery of interventions [25] and at considering such interventions for other populations (adolescents) [24].

Our analysis of the trend of topics has shown that researchers were focused on understanding and gaining insights into cognitive biases and potentially examining the association between cognitive biases and cravings and aggression in the early days. Over the years, there has been an evolution into examining specific unconscious biases, namely, that of attention and approach biases. In the most recent years, the investigations have been more focused on examining bias modification/retraining. This is not surprising, given that it is now more likely that cognitive biases are established to be present in various substance disorders and could potentially be modified. While the trend of topics has provided insights into the directions regarding how the field has been progressing, it has not managed to capture some aspects of the work done to date. For example, there has been an interest in examining how technological advances could be utilised in the delivery of cognitive bias modification interventions, and these include the use of technologies such as web and mobile technologies [32,33]. In addition, because of the issues relating to motivation to train, there have been studies exploring the usage of gamification strategies and even codesign methods in improving usability and in ensuring that the interventions cater to the needs of patients [34,35].

There are several strengths of this current study. Firstly, we sampled the Web of Science database, which has been used in several prior bibliometric studies and has been known to continue an extensive collection of potentially relevant citations. Secondly, we applied several filters when screening the articles to ensure that the eventual database comprises relevant articles. Thirdly, we managed to analyse not only the publication trends of the articles, but also identify the changing focus of research within the field. Despite these strengths, there is a limitation. We sampled the database at a specific time point, and there might be articles published more recently that have been missed.

## 5. Conclusions

From our knowledge, this is one of the first bibliometric analysis that has been undertaken to explore all the publications related to cognitive bias in the field of addiction. This review has demonstrated that there have been extensive investigations in this field, with investigations focusing on alcohol, tobacco, and cannabis use disorders; both methods of cognitive bias assessment, that of attention and approach/avoidance biases, have been looked into. The insights gained from this article could inform future research.

**Funding:** M.W.B.Z. is supported by a grant under the Singapore Ministry of Health's National Medical Research Council (grant number NMRC/Fellowship/0048/2017) for Ph.D. training. The funding source was not involved in any part of this project.

**Institutional Review Board Statement:** Not applicable.

**Informed Consent Statement:** Not applicable.

**Data Availability Statement:** All available data have been included in the manuscript.

**Conflicts of Interest:** The author declare no conflict of interest.

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
