# Peer review of "Cognitive Biases and Addictive Disorders: A Bibliometric Review"

_2673-5318, doi:10.3390/psychiatryint3020011_

Round 1
Reviewer 1 Report
Dear Editor, dear Author,
thank you for the opportunity to review this interesting manuscript. The manuscript requires minor revision.
1) Please give a concrete conclusion - mentioning that this is the first article of this type isn't a proper conclusion.
2) please add study flow diagram for better understanding the methodology and results.
3) Please check the references - if they are properly written - some of them have underscript text and some of them are marked with grey color.
Best regards,
Author Response
Dear Reviewer 1,
Thank you for peer reviewing our work. Please find as enclosed my in-line responses to your comments.
- We have expanded the conclusions. The amends are “From our knowledge, this is one of the first bibliometric analysis that has been undertaken to explore all the publications related to cognitive bias in the field of addiction. This review has demonstrated there being extensive investigations in this field, with investigations focusing on alcohol, tobacco and cannabis use disorders; and both methods of cognitive bias assessment, that of attention and approach/avoidance biases have been looked into. From our knowledge, this is the first bibliometric analysis that has been undertaken to explore all the publications related to cognitive bias in the field of addiction. The insights gained from this article could inform future research.”
- We have added an additional Figure (Figure 1) to illustrate the selection process of the articles.
- We have checked the references and amended them accordingly.
Reviewer 2 Report
The article titled "Cognitive Biases & Addictive Disorders: A Biblio-2 metric Review" reviews the investigations into cognitive biases in addictive disorders.
Comments
- The methodolgy should be explained in a better way. A pictographic represenation shall be helpful.
- Figure legends should be more explanatory.
- The results section should be elaborated to explain the outcomes of the analysis.
Author Response
Dear Reviewer 2,
Thank you for peer reviewing our manuscript. Please find our replies to your comments.
- We have, as per Reviewer 1’s comments, included a new flowchart to illustrate the selection of the articles.
- We have made the following changes to the Figure legends, so that they are more self-explanatory.
- Figure 1: A Flowchart of how studies were selected
- Figure 2: Annual Scientific Production of Articles
- Figure 3: Three-field Plot showing the association between countries, authors and keywords used
- Figure 4: Trend of Topics from 2006 through to 2021
- Figure 5: Thematic Evolution of topics over the years
- Figure 6: An Overview of the co-occurrence of authors’ keywords
- We have attempted to include more key phrases at the start of each of the sentences, so that readers could relate the analysis we performed to the results that we have presented.
Reviewer 3 Report
I agree with the authors: The insights gained from this article could inform future research. This review aims to help researchers betther understand the directions of research in the field and identify key research gaps. In other hand I agree with the different strenghts in the study.
The tables and figures used are very clear and accordance with the text. This is an excellent Bibliometric Review; congratulations.
Author Response
Dear Reviewer 3,
We thank you for your kind comments, and your recognition of the importance of our work.
Round 2
Reviewer 2 Report
Accept the revised version